# Clinical Presentation and Mortality of Severe Fever with Thrombocytopenia Syndrome in Japan: A Systematic Review of Case Reports

**DOI:** 10.3390/ijerph19042271

**Published:** 2022-02-17

**Authors:** Kanako Yokomizo, Momoko Tomozane, Chiaki Sano, Ryuichi Ohta

**Affiliations:** 1Faculty of Medicine, Shimane University, Izumo 693-8501, Japan; nyachocoque@gmail.com (K.Y.); sanochi@med.shimane-u.ac.jp (C.S.); 2Department of Postgraduate Medical Education, Japanese Red Cross Society Himeji Hospital, Himeji 670-8540, Japan; pgbo6o5h@gmail.com; 3Community Care, Unnan City Hospital, Unnan 699-1221, Japan

**Keywords:** severe fever with thrombocytopenia syndrome (SFTS), systematic review, Japan, mortality, hospital admission, recognition

## Abstract

Severe fever with thrombocytopenia syndrome (SFTS) is an infection mediated by ticks and has been reported to have a high mortality rate in Japan. At our hospital, we reported three cases of SFTS with relatively positive outcomes. We reviewed reports of SFTS cases in Japan to clarify the current state of the disease in Japan, the treatment provided, and its outcome. The Ichushi Web was searched for literature using the following terms as keywords: “SFTS” or “severe fever with thrombocytopenia syndrome”. Overall, 174 cases were collected and reviewed. The mean age of patients was 70.69 years old, and the mortality rate was 35%. The dead group was significantly older (*p* < 0.001) than the alive group, had a significantly shorter period from symptom onset to hospital admission, and experienced significantly more hemorrhage-related and neurological symptoms. Further, the most frequently provided treatment methods were adrenocorticosteroids, antibiotics, and conservative treatment. The low recognition rate of SFTS in Japan might lead to a misdiagnosis or delay in diagnosis and treatment, especially in mild to moderate cases. Medical professionals and citizens who live in areas inhabited by ticks need to be informed about SFTS to appropriately diagnose and manage SFTS cases in Japan in the future.

## 1. Introduction

Severe fever with thrombocytopenia syndrome (SFTS) is a viral infection transmitted by tick bites. It is often prevalent during spring to summer and is characterized by fever, thrombocytopenia, hemorrhage, and gastrointestinal symptoms. SFTS was first identified in China in 2009 and was first reported in 2011 in humans [1,2]. In 2012, SFTS cases were reported in South Korea and Japan, and the number of patients has been increasing in East Asia and other regions [3,4]. The cumulative total number of reported patients was 7419 in China in 2016, 335 in South Korea in 2016, and 319 in Japan in 2017.

There are large gaps in the literature regarding differences in mortality rates between countries. For example, the reported mortality rates of those diagnosed with SFTS were 4.8%, 21.8%, and 27% in China, South Korea, and Japan, respectively [5,6,7,8]. In Japan, the first case was confirmed in Yamaguchi Prefecture, located in the Chugoku region, in 2013; since then, 40 to 90 cases have been reported annually, although they have been limited to the Shizuoka Prefecture in Western Japan [7]. Of the 303 reported cases in the Infectious Diseases Weekly Report from April 2013 to October 2017, 133 cases were analyzed epidemiologically based on patient information [7,8]. One study showed that the mean age of patients was 74 years, with the majority being 60 years or older; the male-to-female ratio was roughly 1:1, and the mortality rate of all cases during the study period was higher than that of other countries [7].

Diverse studies have been conducted to identify the factors associated with the severity of SFTS. For example, in a prospective study involving 2096 patients with SFTS who were treated in hospitals located in Henan Province, China, from April 2011 to October 2017, the mean age at the time of hospital admission was 61.4 years, 59% of those affected were female, and the mortality rate was 16.2% [9]. It was also found that patients had a higher risk of mortality if they exhibited the following characteristics: male sex, older age, delay from symptom onset to hospital admission, diarrhea, dyspnea, hemorrhage, and neurological symptoms [9]. Further, certain laboratory measures such as higher lactate dehydrogenase (LDH), aspartate aminotransferase (AST), and blood urea nitrogen (BUN) levels, and elevated neutrophil counts were associated with higher mortality rates [9]. The same study also showed that the viral load in blood samples was a strong predictor of fatal outcomes. Similar results have been reported by studies conducted in Japan [7,8]. Therefore, the factors associated with the severity of SFTS need to be examined in each case, and conservative treatment should be provided appropriately because of the lack of evidence related to the use of antiviral drugs and other intensive treatments [7,8].

The severity and mortality rates of SFTS differ considerably for each country. Specifically, the mortality rate in Japan has been shown to be approximately 30%, indicating a poor prognosis [3,7,9]. In contrast, other countries such as China and South Korea have reported lower mortality rates, despite the lack of significant differences in the quality of medical care provided. Further, the rate ranges from low to high among articles published in East Asia, suggesting that there is a wide degree of variability in the data [3]. In particular, the three SFTS cases described in detail in this article who were treated at our hospital had relatively positive outcomes, indicating a discrepancy from previously reported mortality rates.

Thus, we aimed to investigate the treatments most often provided and the clinical outcomes among patients diagnosed with SFTS in Japan, focusing on academic reports published in Japanese. We also aimed to highlight the treatment and outcomes of three specific SFTS cases from rural areas.

## 2. Materials and Methods

### 2.1. Case Series

Of the patients hospitalized at Unnan City Hospital between January 2013 and July 2021, the cases of three patients diagnosed with SFTS were described with regard to symptom onset, diagnosis, and treatment. We have obtained informed consent from each patient.

### 2.2. Systematic Review

We searched the Ichushi Web (http://www.jamas.or.jp/service/ichu/about.html: Access date, 1 August 2021) for literature published between January 2013 and July 2021, using the following terms as keywords: “SFTS” or “severe fever with thrombocytopenia syndrome”. Ichushi Web is an online Japanese literature search system provided by the Non-Profit Japan Medical Abstracts Society. The Ichushi Web database covers about 10 million medical papers from 6000 journals in Japan and is often used to search for Japanese literature. As the inclusion criteria for the literature, we included case reports and studies that targeted humans and were original articles or abstracts presented for academic conferences held in Japan. We excluded studies that were not academic reports, those that targeted other organisms, and papers without original data. The following data were collected from each case report: reported year, region, patient information (age, sex, chief complaint, duration from symptom onset to hospital admission, whether the individual was dead or alive, the type of treatment provided, LDH level (U/L), AST level (U/L), platelet count (×10^4^/μL), hemoglobin level (g/dL), white blood cell count (/μL), neutrophil count, and whether they developed hemorrhage or had neurologic or gastrointestinal symptoms and cytopenia).

In addition, the region was divided into eight areas, which included the following respective prefectures:Hokkaido.Tohoku: “Aomori”, “Iwate”, “Miyagi”, “Akita”, “Yamagata”, “Fukushima”.Kanto: “Ibaraki”, “Tochigi”, “Gunma”, “Saitama”, “Chiba”, “Tokyo”, “Kanagawa”.Chubu: “Niigata”, “Toyama”, “Ishikawa”, “Fukui”, “Yamanashi”, “Nagano”, “Gifu”, “Shizuoka”, “Aichi”.Kinki: “Mie”, “Shiga”, “Kyoto”, “Osaka”, “Hyogo”, “Nara”, “Wakayama”.Chugoku: “Tottori”, “Shimane”, “Okayama”, “Hiroshima”, “Yamaguchi”.Shikoku: “Tokushima”, “Kagawa”, “Ehime”, “Kochi”.Kyushu: “Fukuoka”, “Saga”, “Nagasaki”, “Kumamoto”, “Oita”, “Miyazaki”, “Kagoshima”, “Okinawa”.

Hemorrhagic cases were defined as those in which patients had clinical symptoms such as subcutaneous and gastrointestinal bleeding or if disseminated intravascular coagulation (DIC) was clearly noted. Cases with neurologic symptoms were defined as those in which patients had an indication of staggering or disturbance of consciousness; cases that resulted in death were also included. Gastrointestinal symptoms included nausea, vomiting, diarrhea, and loss of appetite. We defined cases that had a decrease in the counts of at least one blood cell type, including red blood cells, white blood cells, and platelets, as having “cytopenia”. To ensure that the study quality was reliable, we collected and analyzed the data based on the PRISMA standards. This systematic review was registered in PROSPERO (registration number: 287115) and conducted according to the standard procedure.

### 2.3. Statistical Analysis

The cases were categorized into two groups—either dead or alive—to investigate and compare the patients’ demographic and clinical information, including age, sex, chief complaint, duration from symptom onset to hospital admission, type of treatment provided, LDH level (U/L), AST level (U/L), platelet count (×10^4^/μL), hemoglobin level (g/dL), white blood cell count (/μL), neutrophil count, and whether the patient developed hemorrhage or had neurological or gastrointestinal symptoms and cytopenia. Student’s *t*-tests and Mann–Whitney U tests were performed for comparisons of parametric and non-parametric data, respectively. Statistical significance was defined by a *p*-value < 0.05. All statistical analyses were performed using EZR (Saitama Medical Center, Jichi Medical University, Saitama, Japan), a graphical user interface for the R software environment (The R Foundation, Vienna, Austria).

## 3. Results

### 3.1. Case Series

#### 3.1.1. Case 1 (80-Year-Old Female)

The patient had no notable medical history, and health checkups conducted every year had shown no abnormalities. On Day X − 4, she was referred to a dermatologist for eczema on the right side of her neck and was prescribed antibiotics; on the same day, she developed chills and fever symptoms. A high fever of approximately 39 °C persisted for a few days, which caused her to visit our hospital on Day X. Vital signs upon arrival at the hospital were as follows: body temperature, 38.7 °C; blood pressure, 135/72 mmHg; heart rate, 62 beats/min; SpO_2_, 94%; and stable respiration. She was found to have cytopenia of two cell types (white blood cells, 2400/μL; platelets, 7.3 × 10^4^/μL), elevation of liver and muscle enzymes (AST, 80 IU/L; ALT, 29 IU/L; creatine kinase, 364 IU/L), and dehydration (BUN, 29 mg/dL; creatinine, 0.85 mg/dL), which resulted in her being admitted to the hospital on the same day for investigation of the cause of fever. Cytopenia further progressed after hospital admission, and abnormally high levels of ferritin (17,069.5 ng/mL) were observed on Day X + 2; therefore, a bone marrow examination was conducted. As a result, she was diagnosed with hemophagocytic syndrome and treated with adrenocorticosteroids. The patient was finally diagnosed with SFTS using polymerase chain reaction (PCR) testing. She underwent fasting because of disturbance of consciousness, probably due to encephalopathy, and nutrition was managed with a central venous catheter. The platelet count was the lowest at 1.7 × 10^4^/μL on Day X + 5; thereafter, its value and other laboratory data improved after treatment with fluid and platelet transfusions. Although rehabilitation took a long time, the patient was discharged after 1.5 months of hospitalization.

#### 3.1.2. Case 2 (88-Year-Old Female)

The patient had a medical history of atrial fibrillation, hypertension, and hypothyroidism, but led a normal life until Day X − 1. She consulted a nearby doctor because she acutely developed staggering and gait disturbance on the morning of Day X and was referred to our hospital for close examination and treatment. Vital signs upon hospital arrival were as follows: body temperature, 39.7 °C; blood pressure, 138/67 mmHg; heart rate, 87 beats/min; SpO_2_, 96%; and respiration rate, 20 breaths/min. Blood tests revealed that she had cytopenia of three cell types (white blood cells, 2600/μL; red blood cells, 370 × 10^4^/μL; platelets, 9.6 × 10^4^/μL) and a prolonged activated partial thromboplastin time (57.7 s), and was admitted to the hospital for investigation of the cause of fever. Thereafter, abnormally high levels of ferritin (9824.7 ng/mL on Day X + 3) and elevation of AST (maximum at 199 IU/L on Day X + 7) were observed. Bone marrow examination and PCR testing were conducted, and she was diagnosed with hemophagocytic syndrome and SFTS. The platelet count was the lowest at 4.1 × 10^4^/μL on Day X + 5, although it improved thereafter. Treatment with fluid transfusion, enteral nutrition, and adrenocorticosteroids was provided, which improved the symptoms associated with SFTS. However, she repeatedly developed sepsis due to a new bacterial infection (pneumonia and pyelonephritis) and needed to be treated with antibiotics, which required a long period of hospitalization. Finally, the patient was discharged to a long-term care facility after 5 months of hospitalization.

#### 3.1.3. Case 3 (83-Year-Old Female)

The patient had a history of paroxysmal supraventricular tachycardia, dyslipidemia, and glaucoma. She had butchered wild boars that were exterminated daily. In the afternoon on Day X − 3, she developed fever and nausea. The cause of the fever was unknown, and she visited our hospital on Day X − 1. Vital signs at the time of arrival to the hospital on Day X were as follows: body temperature, 38.2 °C; blood pressure, 113/84 mmHg; heart rate, 154 beats/min; SpO_2_, 94%; and respiration rate, 18 breaths/min. The blood test showed cytopenia of two cell types (white blood cells, 2300/μL; platelets, 10.3 × 10^4^/μL) and elevation of a liver enzyme (AST 75 IU/L); however, the C-reactive protein value was just slightly elevated. Due to several small blood spots on the left palm of the patient and episodes of tick bites, SFTS was considered as the cause and was confirmed by PCR testing. After hospital admission, dehydration and prerenal acute kidney injury were observed due to the inability to consume oral fluids, so fluid transfusion was provided to treat the dehydration. The platelet count continued to decrease and was the lowest at 6 × 10^4^/μL on Day X + 5, although it improved thereafter. The level of the liver enzyme AST also peaked and improved after reaching its maximum value of 143 IU/L on Day X + 3. She was able to consume food orally, and no other complications developed. She was discharged on Day X + 14.

### 3.2. Systematic Review

#### 3.2.1. Data Collection Results

We searched the Ichushi Web on 1 August 2021, to extract 160 articles. From these articles, 136 studies targeting humans were extracted. We excluded those without full-text literature and those describing the same case, and the final number of available literature articles was 129. The number of cases included in the available literature was 171. By adding the three cases described in this review series, the total number of cases was 174 (Figure 1).

#### 3.2.2. Demographic Data

The overall mean age of the included cases was 70.69 years (standard deviation = 14.04). The mortality rate was 35.1%. In terms of age, the dead group was significantly older than the alive group (*p* < 0.001). The duration from symptom onset to hospital admission was significantly shorter in the dead group than in the alive group. In addition, the dead group developed hemorrhage and neurological symptoms significantly more frequently. In contrast, the alive group developed cytopenia significantly more frequently. No significant differences were observed in the other factors between the alive and dead groups (Table 1).

The chief complaints at the time of presentation were as follows, in descending order for all cases: fever (71.8%), malaise (30.5%), diarrhea (22.4%), appetite loss (20.1%), nausea or vomiting (13.2%), pain, including muscle, joint, and lower back pain (10.9%), disturbance of consciousness (9.2%), difficulty in moving/walking (6.3%), abdominal pain (6.3%), staggering or weakness (5.7%), chills (5.7%), headaches (5.7%), lymphadenopathy (4.0%), skin rash (3.4%), and gastrointestinal symptoms (not specified) (1.7%).

The alive group experienced malaise significantly more frequently than the dead group (*p* = 0.025), and skin rash was observed significantly more often in the dead group (*p* = 0.021). There were no significant differences between the alive and dead groups for the other symptoms (Table 2).

The most common treatments provided for all cases were as follows, in descending order: adrenocorticosteroid (41.4%), antibiotics (39.1%), conservative treatment, including fluid transfusion (18.4%), dialysis or plasma exchange (13.8%), DIC therapy (10.3%), blood transfusion (10.3%), antiviral drugs (7.5%), immunoglobulins (6.9%), respiratory management, including endotracheal intubation (6.3%), antifungal drugs (6.3%), granulocyte colony stimulating factor (G-CSF) (2.9%), immunosuppressive agents (2.3%), and no treatment or follow-up observation (1.7%).

The numbers of concervative treatment including fluid transfusion, and antiviral drugs used to treat SFTS cases were significantly higher in the alive group (*p* = 0.04, 0.035, respectively). On the other hand, the number of immunoglobulins, and immunosuppressive agents used to treat SFTS cases were significantly higer in the dead group (*p* = 0.035, 0.014, respectively). There was no significant difference between the alive and dead groups for the other treatments (Table 3).

#### 3.2.3. Number of Cases According to Year of Reporting

The first case was reported in 2013, and there were seven reported cases that year. Thereafter, 27 cases were reported in 2014, and a maximum of 36 cases was reported in 2015; the number of reported cases decreased through 2021 (Figure 2).

#### 3.2.4. Number of Cases According to Region

Figure 3 shows the reported case numbers by region and prefecture. No cases were reported in the Hokkaido, Tohoku, and Kanto regions, and all cases were limited to the western area of the Chubu region. The number of cases reported in the Chugoku, Shikoku, and Kyushu regions was 49, 47, and 61, respectively. In these three regions, cases were identified in all prefectures, except for the Tottori prefecture of the Chugoku region. The prefectures that had the most cases were Ehime in the Shikoku region and Kagoshima in the Kyushu region (21 cases each), followed by Hiroshima in the Chugoku region (20 cases), Tokushima in the Shikoku region (16 cases), and Miyazaki in the Kyushu region (13 cases).

## 4. Discussion

The mean age of all cases was 70.69 years, and the mortality rate was approximately 35 percent, which was mostly consistent with those reported in epidemiological surveys conducted in Japan (27%) [8,9]. We found the same tendency of higher mortality rates than in other Asian countries. In comparing the alive and dead groups, the dead group was significantly older than the alive group, had a significantly shorter period from symptom onset to hospital admission, and had significantly more hemorrhagic and neurologic symptoms. The chief complaints were fever, malaise, and gastrointestinal symptoms such as diarrhea, appetite loss, and vomiting. The reported case numbers declined after peaking in 2015, and the cases were limited to the West Japan area, mainly in the Chugoku, Shikoku, and Kyushu regions. In addition, the most frequently provided treatment methods were adrenocorticosteroids, antibiotics, and conservative treatments such as fluid transfusion.

This research revealed that SFTS cases were mainly observed in the West Japan area. Therefore, one should pay attention to the spread of this disease in the future. According to a survey conducted by the Japan National Institute of Infectious Diseases in 2014, ticks carrying the SFTS virus were identified not only in regions where patients were diagnosed with SFTS (Chugoku, Shikoku, and Kyushu regions) but also in regions where there were no confirmed SFTS cases (Hokkaido, Tohoku, Kanto, and Chubu regions), suggesting that the virus-carrying ticks are widespread across Japan [7,8,10,11]. In addition, various tick species are known to spread the SFTS virus. *Haemaphysalis longicornis* as well as *Amblyomma testudinarium*, *Haemaphysalis flava*, *Haemaphysalis formosensis*, *Haemaphysalis hystricis*, *Haemaphysalis kitaokai*, *Haemaphysalis longicornis*, and *Haemaphysalis megaspinosa* can transmit the SFTS virus to humans [7,8,10,11]. As these ticks are prevalent in Asia, one must be cautious of the spread of the virus in various areas in Asia.

The SFTS virus is transmitted by the ticks through a life cycle involving wild animals, including deer, wild boar, cats, and dogs [12,13]. A survey conducted by the Japan National Institute of Infectious Diseases in 2015 reported that 43.2% of deer and 8.6% of wild boars had confirmed SFTS infections based on PCR tests of the saliva or blood in the Yamaguchi prefecture, where the first patient carrying the SFTS virus was identified [8,14]. In addition, as reported by a survey conducted in the area where the patients with SFTS were newly identified in 2014, the prevalence of anti-SFTS virus antibodies in raccoons, raccoon dogs, and cats increased preceding the identification of the index case, suggesting that the SFTS spread between wild animals was a risk factor for SFTS in humans [13,15,16].

A survey by the Ministry of the Environment reported that the habitat area of Japanese deer and wild boars was mainly West Japan, although the area had been spreading northward every year [17,18,19,20]. A previous study concluded that wildlife habitat area changes also accelerated the spread of infection of Japanese spotted fever and scrub typhus, both of which are related to ticks [21,22,23]. The spread of the habitat area of the wild animals may also be associated with the distribution of the ticks carrying the SFTS virus and the differences between regions in terms of case prevalence; therefore, the spread of infection should be studied in the future.

This research showed a higher mortality rate (approximately 30%) for SFTS cases reported in Japan than for those reported in China and South Korea [5,6]. According to a report from China, the mortality rate in China was estimated to be 12.2%, which is much lower than that reported in Japan [24]. It is important to note that the data we collected in this research showed a significantly shorter time from symptom onset to hospital admission in the dead group than in the alive group, although these periods were longer in the reports from China [9]. It is also noted that the mean age in Japan showed in this study, 70.69, was higher than those reported in China, 60.5 [24]. We did not consider the difference in mean age between Japan and South Korea because those in South Korea was unknown [5]. Currently, each prefecture in Japan has been working towards raising awareness of SFTS, and the public has gradually come to recognize the disease [25]. However, it is still insufficient, owing to the limited awareness in the regions where cases occur. Therefore, medical professionals tend to consider the possibility of SFTS less frequently than those in other countries.

Those who developed severe conditions acutely might have received a quick response from clinicians, but cases with mild to moderate symptoms may not have been fully diagnosed. Undiagnosed disease may also be associated with the fact that patients do not complain about the symptoms because of a lack of awareness of the possibility of SFTS [26]. In terms of the cases treated in the community hospital presented in this research, we observed that the patients did not die but achieved remission through conservative treatment provided in the early stage, even though some of them had severe conditions.

The use of antiviral drugs can be considered in critical cases of SFTS. In this study, the numbers of patients received concervative treatment including fluid transfusion, and administered antiviral drugs were significantly higher in the group that was still alive. Previous studies have reported that there was limited evidence supporting active treatment, whereas conservative treatment administered appropriately was the most critical course of action [7,17]. There have been discrepancies between Japan and other Asian countries in diagnosing SFTS and in collecting data related to its prognosis and mortality. In our opinion, this is partly because recognition of SFTS remains low in Japan, and mild to moderate cases may be overlooked. Subsequent studies should investigate the effectiveness of antiviral drugs against SFTS in critically ill patients.

Consistent with the results of the overall systematic review, the three cases treated at our hospital primarily had nonspecific symptoms, such as fever, gastrointestinal symptoms, and malaise. Of those, two cases reported no evident episodes of a tick bite, although they were diagnosed with SFTS based on laboratory test abnormalities, including cytopenia, during hospitalization. It took several days to decide to perform the PCR test for a definitive diagnosis. Many patients with SFTS have relatively mild symptoms [27,28]. If there is no need for hospital admission, those patients might not be tested for SFTS and might be provided with outpatient care for the common cold and other similar diseases. This delay in treatment could be one of the reasons for the higher mortality rate compared to that of other countries. The current primary treatment for SFTS is conservative treatment, and no specific treatment protocols have been established. Considering the further need to collect and study additional cases, we believe it is essential to conduct definitive testing appropriately to prevent overlooking those afflicted with the disease [17]. This may be achieved by conducting detailed interviews and physical examinations, including of the living environment, crusted skin regions, blood spots, and more for cold and gastrointestinal symptoms, especially in the West Japan SFTS area [29]. From the perspective of infection prevention, ticks carrying the SFTS virus are widespread in regions without SFTS cases; therefore, the recognition of SFTS needs to be improved nationwide by conducting surveys for SFTS infection in wildlife and predicting the risk of infection in humans [30,31]. We believe that additional scrutiny is particularly important among people with a high risk of infection, such as those working in the forestry and hunting industry.

This study had some limitations. First, this study only included Japanese literature; thus, racial differences in the findings must be considered. This study focused on the present conditions for the management of SFTS in Japan, and it investigated differences in the prevalence and management of SFTS in an Asian context. Second, most of the investigation targets were case reports; thus, all cases in Japan may not have been identified. In this research, we focused on Japanese SFTS cases by comprehensively utilizing the most famous Japanese search engines to avoid overlooking rare presentations from local conferences and journals in Japan. Future studies should investigate the effectiveness of interventions based on information about the number of confirmed SFTS cases in communities and changes in mortality rates among those affected. These factors would lead to a more realistic understanding of SFTS infections and associated clinical outcomes.

## 5. Conclusions

This study clarified the present conditions of the clinical presentation and mortality of SFTS in Japan through a systematic review of Japanese case reports. The present conditions for the management of SFTS can be negatively impacted by the low cognizance of SFTS, which can lead to misdiagnosis and delayed treatment. In addition, the mortality rate may be high because of overlooked SFTS cases with mild symptoms. In Japan, information regarding SFTS should be disseminated not only among health care professionals, but also among citizens living in places with ticks to ensure the effective diagnosis and management of SFTS in Japan.

## Figures and Tables

**Figure 1 ijerph-19-02271-f001:**
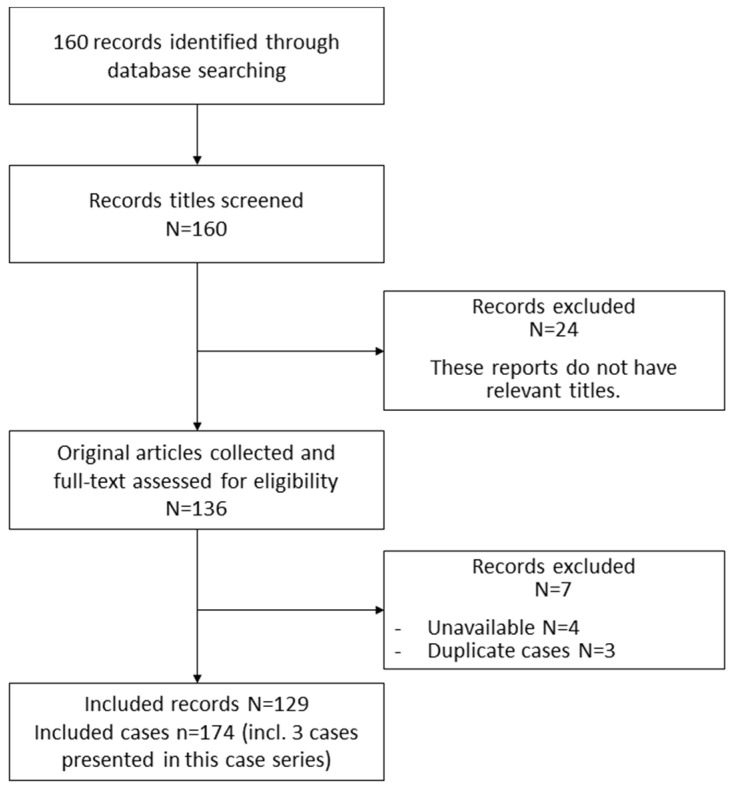
Selection flowchart.

**Figure 2 ijerph-19-02271-f002:**
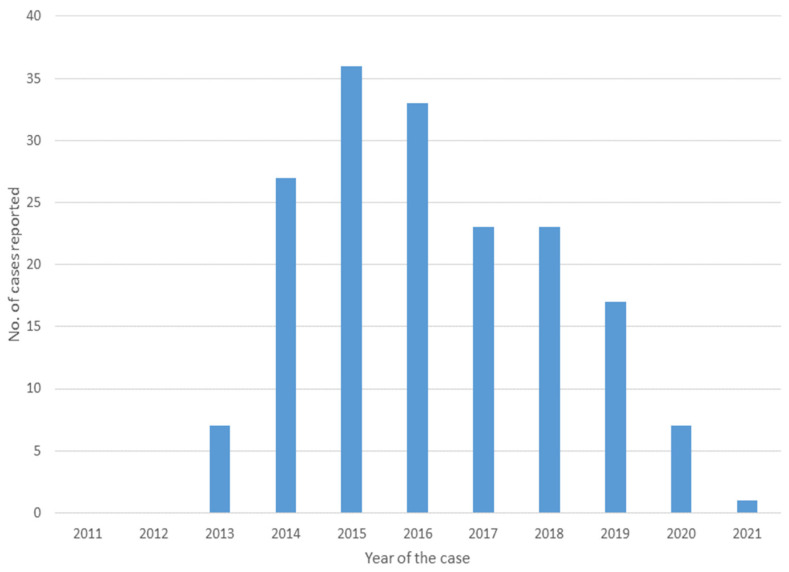
Case reports from 2011 to 2021.

**Figure 3 ijerph-19-02271-f003:**
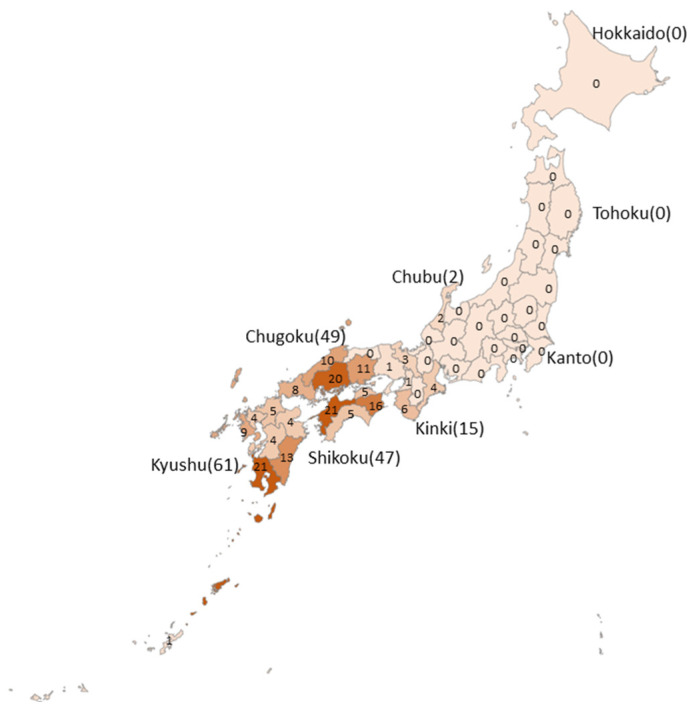
Geographic prevalence of the cases.

**Table 1 ijerph-19-02271-t001:** Demographic data of the included cases.

Factor	Total	Dead	Alive	*p*-Value
n (%)	174	61 (35.06)	113 (64.94)	
Age (SD)	70.69 (14.04)	75.92 (11.06)	67.87 (14.78)	<0.001
Male sex (%)	94 (54.0)	31 (50.8)	63 (55.8)	0.748
Time from symptom onset to hospital admission (SD)	4.71 (2.42)	3.97 (1.94)	5.10 (2.57)	0.034
Symptoms				
Cytopenia (%)	161 (92.5)	52 (85.2)	109 (96.5)	0.013
Gastrointestinal (%)	106 (60.9)	37 (60.7)	69 (61.1)	1
Hemorrhage (%)	49 (28.2)	26 (42.6)	23 (20.4)	0.001
Neurological (%)	113 (64.9)	59 (96.7)	54 (47.8)	<0.001
Laboratory data				
Hemoglobin (SD)	14.23 (1.51)	14.25 (1.43)	14.22 (1.63)	0.955
Aspartate aminotransferase (IQR)	227(23, 4500)	271 (27, 1845)	207(23, 4500)	0.129
Lactate dehydrogenase (IQR)	606(213, 9700)	606 (218, 6785)	577 (213, 9700)	0.738
Platelets (IQR)	6.35 (1.80, 15.70)	6.75 (2.70, 15.70)	5.20 (1.80, 14.20)	0.068
White blood cells (IQR)	1600 (560, 7810)	1730 (600, 7810)	1560 (560, 5300)	0.405
Neutrophil/white blood cell ratio (SD)	60.52 (21.34)	60.02 (28.15)	60.81 (16.86)	0.918

**Table 2 ijerph-19-02271-t002:** Chief complaints among patients diagnosed with SFTS.

	Total	Alive	Dead	*p*-Value
n	174	113	61	
Fever (%)	125 (71.8)	83 (73.5)	42 (68.9)	0.597
Malaise (%)	53 (30.5)	41 (36.3)	12 (19.7)	0.025
Diarrhea (%)	39 (22.4)	25 (22.1)	14 (23.0)	1
Appetite loss (%)	35 (20.1)	22 (19.5)	13 (21.3)	0.844
Nausea, vomiting (%)	23 (13.2)	13 (11.5)	10 (16.4)	0.36
Pain, including muscle, joint, and lower back pain (%)	19 (10.9)	16 (14.2)	3 (4.9)	0.076
Disturbance of consciousness (%)	16 (9.2)	8 (7.1)	8 (13.1)	0.27
Difficulty in moving/walking (%)	11 (6.3)	7 (6.2)	4 (6.6)	1
Abdominal pain (%)	11 (6.3)	10 (8.8)	1 (1.6)	0.099
Staggering, weakness (%)	10 (5.7)	6 (5.3)	4 (6.6)	0.742
Chills (%)	10 (5.7)	6 (5.3)	4 (6.6)	0.742
Headache (%)	10 (5.7)	9 (8.0)	1 (1.6)	0.168
Lymphadenopathy (%)	7 (4.0)	5 (4.4)	2 (3.3)	1
Skin rash (%)	6 (3.4)	1 (0.9)	5 (8.2)	0.021
Gastrointestinal symptoms (not specified) (%)	3 (1.7)	2 (1.8)	1 (1.6)	1
Other (%)	11 (6.3)	8 (7.1)	3 (4.9)	0.749

**Table 3 ijerph-19-02271-t003:** Treatments administered to those diagnosed with SFTS.

	Total	Alive	Dead	*p*-Value
n	174	113	61	
Adrenocorticosteroids (%)	72 (41.4)	44 (38.9)	28 (45.9)	0.421
Antibiotics (%)	68 (39.1)	40 (35.4)	28 (45.9)	0.195
Conservative treatment, including fluid transfusion (%)	32 (18.4)	26 (23.0)	6 (9.8)	0.04
Dialysis, plasma exchange (%)	24 (13.8)	13 (11.5)	11 (18.0)	0.255
Disseminated intravascular coagulation therapy (%)	18 (10.3)	10 (8.8)	8 (13.1)	0.437
Blood transfusion (%)	18 (10.3)	13 (11.5)	5 (8.2)	0.607
Antiviral drugs (%)	13 (7.5)	12 (10.6)	1 (1.6)	0.035
Immunoglobulins (%)	12 (6.9)	4 (3.5)	8 (13.1)	0.026
Respiratory management, including endotracheal intubation (%)	11 (6.3)	4 (3.5)	7 (11.5)	0.052
Antifungal drugs (%)	11 (6.3)	7 (6.2)	4 (6.6)	1
Colony-stimulating factor (%)	5 (2.9)	5 (4.4)	0 (0.0)	0.164
Immunosuppressive agents (%)	4 (2.3)	0 (0.0)	4 (6.6)	0.014
No treatment, follow-up observation (%)	3 (1.7)	3 (2.7)	0 (0.0)	0.553
Other (%)	8 (4.6)	3 (2.7)	5 (8.2)	0.13

## Data Availability

All relevant datasets in this study are presented in the manuscript.

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
