# Peer review of "Clinical Presentation and Mortality of Severe Fever with Thrombocytopenia Syndrome in Japan: A Systematic Review of Case Reports"

_ijerph, 2022, doi:10.3390/ijerph19042271_

Round 1
Reviewer 1 Report
In this paper, the authors presented three cases of SFTS they treated and reviewed the case reports of SFTS written in Japanese in this paper. There are no reports of as many as 174 cases with SFTS in Japan, and this study may be valuable. However, some mistakes are noticeable in the submitted manuscript. In Table 3, the total number of cases receiving antiviral drugs does not match the sum of those in the Alive group and in the Death group. The authors made the same mistakes in the number of cases that underwent conservative treatment, blood transfusion, G-CSF and no treatment. There are some inappropriate and inadequate references [e.g., the reference 9 (page 2, line 46) and the reference 11 (page 2, line 55) are not appropriate for their respective description.).
Author Response
Responses to the reviewers’ comments
Thank you very much for reviewing our manuscript and providing suggestions for its improvement. We have provided point-by-point responses to the reviewers’ comments below; our revisions are indicated in red font in the main document. We hope that the revised manuscript meets the journal’s requirements and can now be considered for publication.
In this paper, the authors presented three cases of SFTS they treated and reviewed the case reports of SFTS written in Japanese in this paper. There are no reports of as many as 174 cases with SFTS in Japan, and this study may be valuable. However, some mistakes are noticeable in the submitted manuscript.
In Table 3, the total number of cases receiving antiviral drugs does not match the sum of those in the Alive group and in the Death group.
Response:
Thank you for your valuable feedback. We have reviewed the data shown in Table 3 and have revised the numbers they added up correctly and were consistent.
The authors made the same mistakes in the number of cases that underwent conservative treatment, blood transfusion, G-CSF and no treatment.
Response:
Thank you for your comment. As mentioned above, we have reviewed the data in Table 3 and have adjusted the numbers accordingly.
There are some inappropriate and inadequate references [e.g., the reference 9 (page 2, line 46) and the reference 11 (page 2, line 55) are not appropriate for their respective description.).
Response:
Thank you for noting the issues with some of the references. Per your comment, we have omitted references 9 and 11 from the article; the rest have been renumbered accordingly.
Reviewer 2 Report
In the article “Clinical Presentation and Mortality of Severe Fever with Thrombocytopenia Syndrome in Japan: A Systematic Review of Case Reports”, the authors describe the case histories of three severe fever with thrombocytopenia syndrome (SFTS) cases diagnosed at Unnan City Hospital, Japan from 2013–2021, and performed a systematic review of 174 SFTS cases diagnosed in Japan from 2013–2021. The newly reported case histories were informative, and the systematic review was performed according to established guidelines. Given the low recognition of SFTS in Japan highlighted by the authors, this article will be important for increasing SFTS awareness of both citizens inhabiting and clinicians working in SFTS high-risk areas. Listed below are suggested revisions.
Specific Comments
Line 34: Replace “emerging infectious disease” with “SFTS” to enhance clarity.
Lines 53-55: Please include whether an increase or decrease in each laboratory test parameter was associated with a higher mortality rate.
Lines 55 and 56: What sample type was used to measure viral loads? Blood? Please include this information.
Line 58: Please briefly explain why conservative treatment is recommended over aggressive treatment for SFTS.
Line 79: Delete “the” before “informed”.
Lines 97-104: Can you please a “/” instead of a “,” between the first and last prefecture of each area to correct the line formatting.
Line 106: Change “Hemorrhage” to “Hemorrhagic”.
Line 111: Change “least in” to “at least”.
Lines 122 and 123: Change “was” to “were”.
Line 144: “Polymerase Chain Reaction” should be written in lower-case type.
Line 225: “Granulocyte Colony Stimulating Factor” should be written in lower-case type.
Line 227: Please insert “used to treat SFTS cases were” in the following sentence to enhance clarity: “The number of antiviral drugs used to treat SFTS cases were significantly higher in the live group (P=0.044).”
Line 235: Please replace “the reported number peaked to decrease in 2021” with “the number of reported cases decreased through 2021.”
Line 256: Replace “hemorrhage” with “hemorrhagic”.
Line 258: Replace “number” with “numbers”.
Line 258: Replace “a peak” with “peaking”.
Line 267: Replace “the region” with “regions”.
Lines 269-271: Please list the tick species from which SFTSV has been detected from in Japan. Has SFTSV been detected in ticks other than Haemaphysalis longicornis in Japan?
Line 271: Delete “spreading infectious diseases”.
Lines 271-273: Please clarify/elaborate on whether the deer and wild boars were actively infected with SFTSV (i.e., SFTSV PCR or virus isolation positive) or had evidence of a past SFTSV infection (SFTSV-specific antibodies by serology)? If the animals were actively infected with SFTSV, what type of sample(s) tested positive (e.g., blood)?
Lines 300-303: Can you comment on the use of antiviral drugs versus conservative therapy? Both the text and table 3 report that the number of antiviral drugs was significantly higher in the live group (P=0.044). Is there any evidence to support that treatment with a particular antiviral leads to a better outcome (i.e., survival)?
Lines 336 and 337: This sentence is difficult to understand. Please clarify what your next step is.
Author Response
Responses to the reviewers’ comments
Thank you very much for reviewing our manuscript and providing suggestions for its improvement. We have provided point-by-point responses to the reviewers’ comments below; our revisions are indicated in red font in the main document. We hope that the revised manuscript meets the journal’s requirements and can now be considered for publication.
In the article “Clinical Presentation and Mortality of Severe Fever with Thrombocytopenia Syndrome in Japan: A Systematic Review of Case Reports”, the authors describe the case histories of three severe fever with thrombocytopenia syndrome (SFTS) cases diagnosed at Unnan City Hospital, Japan from 2013–2021, and performed a systematic review of 174 SFTS cases diagnosed in Japan from 2013–2021. The newly reported case histories were informative, and the systematic review was performed according to established guidelines. Given the low recognition of SFTS in Japan highlighted by the authors, this article will be important for increasing SFTS awareness of both citizens inhabiting and clinicians working in SFTS high-risk areas. Listed below are suggested revisions.
Specific Comments
Line 34: Replace “emerging infectious disease” with “SFTS” to enhance clarity.
Response:
Thank you for your valuable feedback. We have replaced “emerging infectious disease” with “SFTS” as suggested.
Lines 53-55: Please include whether an increase or decrease in each laboratory test parameter was associated with a higher mortality rate.
Response:
Thank you for your comment. we have revised the text on lines 56 to 60 to indicate which test parameters were associated with higher mortality rates as follows:
“Further, certain laboratory measures such as higher lactate dehydrogenase (LDH), aspartate aminotransferase (AST), and blood urea nitrogen (BUN) levels, and an elevated neutrophil rate were associated with higher mortality rates [9]. The same study also showed that the viral load in blood samples was a strong predictor of fatal outcomes.”
Lines 55 and 56: What sample type was used to measure viral loads? Blood? Please include this information.
Response:
Thank you for your comment. We apologize that the sample type used to measure the viral loads was unclear. We have revised the text on the indicated lines and added the word “blood” to make the sample type clearer (lines 59 to 60).
“The same study also showed that the viral load in blood samples was a strong predictor of fatal outcomes.”
Line 58: Please briefly explain why conservative treatment is recommended over aggressive treatment for SFTS.
Response:
Thank you for your valuable feedback. We have revised the text to make the reason for choosing conservative over aggressive treatments clearer (lines 61 to 64) as follows:
“Therefore, the factors associated with the severity of SFTS need to be examined in each case, and conservative treatment should be provided appropriately because of the lack of evidence related to the use of antiviral drugs and other intenstive treatments [7,8].”
Line 79: Delete “the” before “informed”.
Response:
Thank you for your comment. As suggested, we have removed the word “the” before “informed consent” (line 82).
Lines 97-104: Can you please a “/” instead of a “,” between the first and last prefecture of each area to correct the line formatting.
Response:
We appreciate your valuable feedback. We have revised the list of prefectures accordingly (lines 102 to 110).
Line 106: Change “Hemorrhage” to “Hemorrhagic”.
Response:
We have changed “Hemorrhage” to “Hemorrhagic” as suggested (line 112).
Line 111: Change “least in” to “at least”.
Response:
We have replaced “least in” with “at least” as suggested (line 118).
Lines 122 and 123: Change “was” to “were”.
Response:
Thank you for your valuable feedback. We have made the suggested change.
Line 225: “Granulocyte Colony Stimulating Factor” should be written in lower-case type.
Response:
Thank you for your comment. We have ensured that the term is written in lower-case type in the main text (line 235) and in in Table 3.
Line 227: Please insert “used to treat SFTS cases were” in the following sentence to enhance clarity: “The number of antiviral drugs used to treat SFTS cases were significantly higher in the live group (P=0.044).”
Response:
We have revised the phrasing related to the use of antiviral drugs as suggested (lines 237 to 238):
"The number of antiviral drugs used to treat SFTS cases was significantly higher in the alive group (P=0.044).”
Line 235: Please replace “the reported number peaked to decrease in 2021” with “the number of reported cases decreased through 2021.”
Response:
As requested, we have revised the phrase about the number of reported cases as follows (line 245):
“Thereafter, 27 cases were reported in 2014, and a maximum of 36 cases was reported in 2015; the number of reported cases decreased through 2021 (Figure 2).”
Line 256: Replace “hemorrhage” with “hemorrhagic”.
Response:
Thank you for your valuable feedback. We have replaced the word “hemorrhage” with “hemorrhagic” as suggested (line 266).
Line 258: Replace “number” with “numbers”.
Response:
We have pluralized the word as you suggested (line 268):
“The reported case numbers declined after peaking in 2015.”
Line 258: Replace “a peak” with “peaking”.
Response:
Thank you for your valuable feedback. We have replaced the words “a peak” with “peaking” (line 268):
“The reported case numbers declined after peaking in 2015.”
Line 267: Replace “the region” with “regions”.
Response:
Thank you for your comment. We have made the suggested change in the revised text on lines 274 to 279:
“According to a survey conducted by the Japan National Institute of Infectious Diseases in 2014, ticks carrying the SFTS virus were identified not only in regions where patients were diagnosed with SFTS (Chugoku, Shikoku, and Kyusyu regions) but also in regions where there were no confirmed SFTS cases (Hokkaido, Tohoku, Kanto, and Chubu regions), suggesting that the virus-carrying ticks are widespread across Japan [7,8,10,11].”
Lines 269-271: Please list the tick species from which SFTSV has been detected from in Japan. Has SFTSV been detected in ticks other than Haemaphysalis longicornis in Japan?
Response:
Thank you for your valuable feedback. Yes, the SFTS virus has been detected in other tick species in Japan. We have added information regarding the names of these ticks in the Discussion section as follows (line 279 to 284):
“In addition, various tick species are known to spread the SFTS virus. Haemaphysalis longicornis as well as Amblyomma testudinarium, Haemaphysalis flava, Haemaphysalis formosensis, Haemaphysalis hystritis, Haemaphysalis kitaokai, Haemaphysalis longicornis, and Haemaphysalis megaspinosa can transmit the SFTS virus to humans [7,8,10,11]. As these ticks are prevalent in Asia, one must be cautious of the spread of the virus in various areas in Asia.”
Line 271: Delete “spreading infectious diseases”.
Response:
Thank you for your comment. We have deleted the phrase “spreading infectious disease” as requested.
Lines 271-273: Please clarify/elaborate on whether the deer and wild boars were actively infected with SFTSV (i.e., SFTSV PCR or virus isolation positive) or had evidence of a past SFTSV infection (SFTSV-specific antibodies by serology)? If the animals were actively infected with SFTSV, what type of sample(s) tested positive (e.g., blood)?
Response:
We appreciate the thoughtful comment We have revised a portion of the Discussion by adding information concerning the confirmation of SFTS infections among animals as follows (lines 286 to 289):
“A survey conducted by the Japan National Institute of Infectious Diseases in 2015 reported that 43.2% of deer and 8.6% of wild boars had confirmed SFTS infections based on PCR tests of the saliva or blood in the Yamaguchi prefecture, where the first patient carrying the SFTS virus was identified [8,14].”
Lines 300-303: Can you comment on the use of antiviral drugs versus conservative therapy? Both the text and table 3 report that the number of antiviral drugs was significantly higher in the live group (P=0.044). Is there any evidence to support that treatment with a particular antiviral leads to a better outcome (i.e., survival)?
Response:
Thank you for your feedback. Based on your suggestion, we have added text to the Discussion section describing the possible effectiveness of antiviral drugs in such cases and added some sentences describing some future directions as follows (lines 321 to 329):
“The use of antiviral drugs can be considered in critical cases of SFTS. In this study, the number of patients administered antiviral drugs was significantly higher in the group that was still alive. Previous studies have reported that there was limited evidence supporting active treatment, whereas conservative treatment administered appropriately was the most critical course of action [7,17]. There have been discrepancies between Japan and other Asian countries in diagnosing SFTS and in collecting data related to its prognosis and mortality. In our opinion, this is partly because recognition of SFTS remains low in Japan, and mild to moderate cases may be overlooked. Subsequent studies should investigate the effectiveness of antiviral drugs against SFTS in critically ill patients.”
Lines 336 and 337: This sentence is difficult to understand. Please clarify what your next step is.
Response:
We apologize those portions of the text were unclear. We have revised the text describing our next steps as follows (lines 358 to 361):
“Future studies should investigate the effectiveness of interventions based on information about the number of confirmed SFTS cases in communities and changes in mortality rates among those affected. These factors would lead to a more realistic understanding of SFTS infections and the associated clinical outcomes.”
Reviewer 3 Report
1. In lines 60-63, the presentation of the severity and mortality rate of SFTS may need to be rearranged.
- In line 70, the clarity of the presentation needs to be improved.
- In lines 82 and 109, rephrasing and explanations of the method will be more reader-friendly.
- The authors stated results in line 142. However, more details are needed so that the reader can understand the impact of the information.
Author Response
Responses to the reviewers’ comments
Thank you very much for reviewing our manuscript and providing suggestions for its improvement. We have provided point-by-point responses to the reviewers’ comments below; our revisions are indicated in red font in the main document. We hope that the revised manuscript meets the journal’s requirements and can now be considered for publication.
- In lines 60-63, the presentation of the severity and mortality rate of SFTS may need to be rearranged.
Response:
Thank you for your valuable feedback. We have comprehensively revised the text pertaining to the severity and mortality of SFTS as follows (lines 65 to 73):
“The severity and mortality rates of SFTS differ considerably for each country. Specifically, the mortality rate in Japan has been shown to be approximately 30%, indicating a poor prognosis [3,7,9]. In constrast, other countiries such as China and South Korea have reported lower mortality rates, despite the lack of significant diffences in the quality of medical care provided. Further, the rate ranges from low to high among articles published in East Asia, suggesting that there is a wide degree of variability in the data [3]. In particular, the three SFTS cases described in detail in this article who were treated at our hospital had relatively positive outcomes, indicating a discrepancy from previously reported mortality rates.“
- In line 70, the clarity of the presentation needs to be improved.
Response:
We apologize that the text at the end of the Introduction was unclear. In addition to the altered text mentioned in the previous response, we have revised the final paragraph of the Introduction to make the aims clearer as follows (lines 74 to 77):
“Thus, we aimed to investigate the treatments most often provided and the clinical outcomes among patients diagnosed with SFTS in Japan, focusing on academic reports published in Japanese. We also aimed to highlight the treatment and outcomes of three specific SFTS cases from rural areas.”
- In lines 82 and 109, rephrasing and explanations of the method will be more reader-friendly.
Response:
Thank you for your valuable feedback. As suggested, we have revised large portions of text in the Methods section for clarity. The list of areas and prefectures has been revised, as has the subsection describing the statistical analyses, which are quoted, respectively, below:
Lines 102 to 111:
- Hokkaido
- Tohoku: “Aomori” “Iwate” “Miyagi” “Akita” “Yamagata” “Fukushima”
- Kanto: ”Ibaraki” “Tochigi” “Gunma” “Saitama” “Chiba” “Tokyo” “Kanagawa”
- Chubu: “Niigata” “Toyama” “Ishikawa” “Fukui” “Yamanashi” “Nagano” “Gifu” “Shizuoka” “Aichi”
- Kinki: “Mie” “Shiga” “Kyoto” “Osaka” “Hyogo” “Nara” “Wakayama”
- Chugoku: “Tottori” “Shimane” “Okayama” “Hiroshima” “Yamaguchi”
- Shikoku: “Tokushima” “Kagawa” “Ehime” “Kochi”
- Kyushu: “Fukuoka” “Saga” “Nagasaki” “Kumamoto” “Oita” “Miyazaki” “Kagoshima” “Okinawa. “
Lines 124 to 134:
“The cases were categorized into two groups, either dead or alive, to investigate and compare the patients’ demographic and clinical information, including age, sex, chief complaint, duration from symptom onset to hospital admission, type of treatment provided, LDH level (U/L), AST level (U/L), platelet count (×104/μL), hemoglobin level (g/dL), white blood cell count (/μL), neutrophil count, whether the patient developed hemorrhage or had neurological or gastrointestinal symptoms and cytopenia. Student’s t-tests and Mann-Whitney U tests were performed for comparisons of parametric and non-parametric data, respectively. Statistical significance was defined by a P-value < 0.05. All statistical analyses were performed using EZR (Saitama Medical Center, Jichi Medical University, Saitama, Japan), a graphical user interface for the R software environment (The R Foundation, Vienna, Austria).”
- The authors stated results in line 142. However, more details are needed so that the reader can understand the impact of the information.
Response:
Thank you for your valuable feedback. As suggested, we have comprehensively revised portions of the Results and Discussion sections to make the impact of the study clearer.
Round 2
Reviewer 1 Report
The authors have revised some points, but some remain uncorrected. In Table 3, the total number of cases receiving antiviral drugs does not match the sum of those in the Alive group and in the Death group. This is the item I pointed out last time. The authors made the same mistakes in the number of male sex in Table 1.
I regret to inform you that your manuscript lacks the accuracy of statistical analysis of the results. Please consider carefully revising this paper and then submit it as a new one.
In doing so, it will be better if you also consider the following points.
・Since this study collects case reports, unreported cases are not reflected in the result. Therefore, this study may include more severe cases than it actually is. The mortality rate in this study was 35%, which is higher than previously reported in Japan at 27%. This point should be mentioned in the limitation section.
・Regarding the high mortality rate in Japan compared to China and South Korea, the authors should consider the difference in patient age in each country.